# The Impact of Vegan and Vegetarian Diets on Physical Performance and Molecular Signaling in Skeletal Muscle

**DOI:** 10.3390/nu13113884

**Published:** 2021-10-29

**Authors:** Alexander Pohl, Frederik Schünemann, Käthe Bersiner, Sebastian Gehlert

**Affiliations:** 1Department of Biosciences of Sport Science, Institute of Sport Science, University of Hildesheim, 31141 Hildesheim, Germany; schue011@uni-hildesheim.de (F.S.); bersin@uni-hildesheim.de (K.B.); gehlert@dshs-koeln.de (S.G.); 2Department for Molecular and Cellular Sports Medicine, German Sports University Cologne, 50933 Cologne, Germany

**Keywords:** vegetarian, vegan, diet, performance, endurance, strength, microbiome, molecular, signaling

## Abstract

Muscular adaptations can be triggered by exercise and diet. As vegan and vegetarian diets differ in nutrient composition compared to an omnivorous diet, a change in dietary regimen might alter physiological responses to physical exercise and influence physical performance. Mitochondria abundance, muscle capillary density, hemoglobin concentration, endothelial function, functional heart morphology and availability of carbohydrates affect endurance performance and can be influenced by diet. Based on these factors, a vegan and vegetarian diet possesses potentially advantageous properties for endurance performance. Properties of the contractile elements, muscle protein synthesis, the neuromuscular system and phosphagen availability affect strength performance and can also be influenced by diet. However, a vegan and vegetarian diet possesses potentially disadvantageous properties for strength performance. Current research has failed to demonstrate consistent differences of performance between diets but a trend towards improved performance after vegetarian and vegan diets for both endurance and strength exercise has been shown. Importantly, diet alters molecular signaling via leucine, creatine, DHA and EPA that directly modulates skeletal muscle adaptation. By changing the gut microbiome, diet can modulate signaling through the production of SFCA.

## 1. Introduction

In recent years, vegetarian and vegan diets and their impact on health and performance have been brought into focus of scientific research. It is well known that nutrition influences exercise performance [1]. Yet, while the relationship of nutrition in general, and also on aspects of performance and adaptation to exercise is well established [2], research on vegetarian and vegan diets and their impact on performance and training adaptation is scarce. The purpose of this review is firstly to summarize the published research on vegetarian and vegan diets with a special emphasis on strength- and endurance-related exercise performance. Secondly, we also aim to highlight the potential impact of those diets on systemic and molecular muscle adaptations through training. In order to be considered as suitable for the first purpose, research items had to meet two criteria. First, subjects in the involved studies had to follow a vegetarian or vegan diet and second, performance outcome had to be measured. Additionally, research on general aspects and properties of endurance and strength performance as well as research that focused on the adaptation of molecular mechanisms affected by those diets was included.

## 2. Properties of Vegetarian and Vegan Diets

Vegetarian diets can be divided into six different types, as shown in Table 1. These different types are depended on the inclusion and exclusion of food sources. Vegan diets show a considerable variety with the omittance of non-rooted vegetables being a classic variation. However, to our knowledge no scientific data show differences in nutritional properties of these variations and their impact on health or performance.

### 2.1. Differences in Macronutrients between Diets

Due to dietary-based food compositions, energy, macronutrient and micronutrient intake vary between diets [3]. Vegan (stated as strict vegetarian in the original article) and omnivorous (stated as non-vegetarian in the original article) diets usually offer the greatest difference concerning macronutrient intake [3]. Vegan diets are usually higher in carbohydrates and fiber, but lower in fat compared to omnivorous and vegetarian diets [3]. There are no significant differences in unsaturated fatty acid intake between diet regimens, although a tendency of a higher mono-unsaturated fatty acid intake in omnivores has been recognized [3]. Vegans consume significantly fewer saturated fatty acids (SFA) and unsaturated docosahexaenoic acid (DHA) compared to omnivores and similar to vegetarians [3]. Protein intake of vegans is slightly lower compared to omnivores but similar to semi vegetarians and lacto-ovo vegetarians, with omnivores consuming the highest amount of animal protein compared to other dietary regimens (see Table 2) [3].

### 2.2. Differences in Micronutrients between Diets

Furthermore, there are differences in micronutrient intake, as vegans consume significantly less vitamin D than omnivores (*p* < 0.05) but not than vegetarians, while none of the examined dietary groups (omnivores, semi-vegetarian, pesco-vegetarian, lacto-ovo-vegetarian, vegan) [3] consumed the daily recommended intake of 600 IU (15 µg) [5,6] (see Table 2). The degree of adherence to a plant-based Mediterranean diet, was found to be positively associated with high circulating levels of vitamin D [7] emphasizing the long term benefits of this diet for the elevation of circulating vitamin D levels.

However, despite the various beneficial aspects of a Mediterranean diet [8] it is not necessarily superior to vegan or omnivorous diets in terms of vitamin D blood levels. It has been shown that omnivores and vegans show higher blood levels of 26.1 ng/mL and 31.6 ng/mL, respectively, compared to 23.0 ng/mL in subjects consuming a Mediterranean diet [7,9].

Magnesium intake is significantly higher (*p* < 0.05) in vegans but not vegetarians compared to omnivores [3] with all groups meeting the daily recommended intake (females: 310–320 mg/day; males: 400–420 mg/day) [10]. This nutrient distribution can also be found in European populations [11], with a more pronounced deficiency in vitamin D intake.

Total creatine concentration measured in skeletal muscle tissue differs between vegetarians and omnivores, with omnivores showing the highest total creatine concentrations [12]. As the body synthesizes approximately 1 g per day of creatine endogenously, food, in the form of meat, fish and poultry, provides an additional 1 g [13]. Due to the restrictive dietary pattern, vegetarians and vegans consume less dietary creatine than omnivores [14] and vegans’ repletion of creatine stores entirely depends on endogenous synthesis [15].

Despite these minor differences in nutritional composition, vegan and vegetarian diets have been shown to be nutritionally adequate in terms of meeting the recommended energy, macronutrient and micronutrient intake, when organized appropriately [1,16,17,18,19].

An appropriately planned vegetarian and vegan diet includes a variety of plant foods [16], however, the supplementation of micronutrients such as vitamin D, vitamin B12 and iron is frequently observed [19].

## 3. Do Vegetarian and Vegan Diets Affect Exercise Performance

The impact of nutrition on exercise performance is well studied. Over the past two decades, research papers on nutrition and exercise performance have rapidly increased in number, peaking in 2020 with 1758 published research items containing the keywords nutrition and exercise performance (source: PubMed, 28.July.2021). However, research on vegetarian and vegan diets and their impact on exercise performance is scarce—only three and six research items, respectively, were published on these topics in 2020 (source: PubMed, 28.July.2021; keywords: vegetarian diet and exercise performance; vegan diet and exercise performance). Due to the limited research in this field, this review takes both, the impact of vegetarian and also vegan diets on exercise performance into account and extracts the essential data of these papers. Research from 1999 to 2021 was examined and 14 research items were identified as suitable for the purpose of this brief narrative review and are summarized in Table 3. These studies are described in detail in the section on vegetarian and vegan diets and endurance performance and vegetarian and vegan diets and strength performance.

## 4. Vegan and Vegetarian Diet and Endurance Performance

### 4.1. Factors That May Affect Endurance Performance Differently between Diets

Endurance performance is usually assessed with the measurement of VO_2_max [20]. It is a common indicator for systemic training effects on global oxidative capacity [20], although endurance performance depends on different physiological subsystems, e.g., mitochondrial abundance and muscle capillary density [20,21,22]. No significant difference in mitochondrial density between vegans and omnivores has been detected, although there was a trend towards a higher relative mitochondrial DNA content (relative amount of mitochondrial DNA to nuclear DNA) in vegans [23]. To our knowledge, no research on vegan or vegetarian diet and capillarization has been conducted yet, but it has been shown that in vitro, the isoflavone *Genistein* from the soybean inhibits neovascularization in bovine microvascular endothelial cells [24]. As vegetarians and vegans consume significantly more soy protein (*p* < 0.05) [3], these diets may influence capillarization. In trained athletes however, VO_2_max critically depends on the cardiac output in combination with the oxygen carrying capacity of the blood and thus hemoglobin concentration [25,26]. The former may be affected by a vegan diet as it positively influences both morphological and functional heart remodeling such as lower relative wall thickness (RWT), and better left ventricular systolic and diastolic function [27]. The RWT describes the relation of wall thickness to chamber dimension [28]. The positive changes in systolic and diastolic function may occur because of the antioxidant properties of vegan and vegetarian diets and an improved endothelial function in vegans and vegetarians [29,30,31]. Moreover, the lower intake of saturated fatty acids (SFA) may be responsible for the slightly better diastolic function in vegans [27].

A healthy adult has a hemoglobin concentration of 12–16 g/dL [32,33] and iron intake has been shown to be a critical component for the maintenance of hemoglobin concentration in endurance athletes [34] but also in vegans and vegetarians [35,36]. Despite the similar dietary iron intake of omnivores, vegetarians and vegans (see Table 2), endurance performance can be influenced by the dietary choice due to the greater bioavailability of animal-derived heme-iron (15–35% absorption) compared to plant-derived non-heme-iron (2–20% absorption) [37]. It has been shown that both, vegans and vegetarians, exhibit a higher prevalence of decreased iron status compared to omnivores which leading to an insufficient hemoglobin synthesis, which can negatively affect endurance performance [26].

Another nutritional factor that may affect endurance performance between diets is vitamin D intake. In subjects with low serum 25-hydroxy vitamin D (25(OH)D) levels, a low phosphocreatine (PCr)/inorganic phosphate (Pi) ratio was observed, suggesting a reduced oxidative phosphorylation in muscles [38]. It has also been reported that a supplementation of vitamin D improved post-exercise PCr and ADP recovery, increased the PCr/Pi ratio, and reduced Pi/adenosine triphosphate (ATP) ratio significantly, proposing an improved mitochondrial oxidative capacity [38,39].

The knockdown of vitamin D receptor (VDR) in C2C12 myoblasts resulted in decreased mitochondrial oxidative capacity and ATP production, further strengthening the role of vitamin D in endurance performance [40]. As vegans consume significantly less vitamin D compared to omnivores [3], this may affect endurance performance. A recent study has shown a positive association between vitamin D status and endurance performance but also showed that vitamin D supplementation did not improve exercise performance [41]. Therefore, data on vitamin D supplementation and endurance exercise performance are still inconsistent and this field requires further research.

Muscular carnosine content may also influence exercise performance. Carnosine is a dipeptide composed of β-alanine and L-histidine [42] and its major physiological functions include muscular pH-buffering and the activation of muscle ATPase to provide energy [43]. As it is highly abundant in beef and absent from plants [43], dietary choices can influence carnosine levels in the long-term [44] and therefore may affect exercise performance [45,46,47]. Carnosine may also influence strength performance [48].

In summary, the properties of vegetarian and vegan diets may have an impact on cardiac output, hemoglobin concentration, mitochondrial function and pH-buffering capacity, possibly affecting endurance performance.

### 4.2. Differences in Substrate Availability between Vegan or Vegetarian and Omnivorous Diets May Affect Endurance Performance

Of major importance for acutely conducted endurance exercise is the substrate availability of the macronutrients fat and carbohydrates [49]. Carbohydrates become the predominant energy source when exercising with intensities of more than 60% of the VO_2_max [50,51]. Endurance exercise carried out with lower intensities relies to a higher degree on fat oxidation [49]. Hence, with increasing exercise intensity, muscle glycogen and plasma glucose oxidation increase whereas fat oxidation declines [52]. These results underpin the essential role of carbohydrates as a fuel for acute endurance performance [49].

As displayed in Table 2, vegans and vegetarians consume 16% and 7% more carbohydrates than omnivores, respectively, which could lead to an advantage in endurance performance.

When performing endurance exercise near VO_2_max, over 80% of energy is mainly supplied from glycogen granules from the intercellular substrate stores of muscle fibers (IMG) [53]. Mitochondrial ATP generation due to carbohydrate metabolism depends mainly on IMG stores as only 20-30% of the fuels are supplied via the capillaries [53] from the blood stream.

However, to our knowledge, no studies investigating the basal density of muscle glycogen granules and intramuscular lipid droplets comparing omnivores, vegetarians or vegans have been conducted yet. Therefore, the advantage of higher basal carbohydrate consumption in vegans or vegetarians towards enhanced endurance performance are not clear. It must be considered that exercise training per se is the most important driver for increasing intramuscular substrate stores [54].

Moreover, the carbohydrate consumption during endurance exercise affects endurance performance [55]. During prolonged endurance exercise and under conditions when IMG stores decline, an increasing amount of glucose is delivered via the blood stream towards working muscle [56]. It is well established that carbohydrate consumption during prolonged endurance exercises extends time to exhaustion [57]. However, the huge amount of sports-nutrition available for acute provision of carbohydrates is mainly plant-based [58] and to date there is no scientific evidence that pure vegan compared to vegetarian and omnivorous energy sources (power gels, energy bars, isotonic carbohydrate drinks) show functional differences in gastrointestinal emptying, carbohydrate availability or other factors that may affect endurance performance.

In conclusion, endurance performance is affected on multiple levels. While exercise dominantly stimulates endurance exercise adaptation, different macro- and micronutrient intake between diets may affect cardiac output, oxygen carrying capacity, mitochondrial function and substrate availability. It has yet to be determined how diets impact endurance exercise capacity (Figure 1).

### 4.3. Evidences on Vegetarian and Vegan Diets and Endurance Performance

Nine studies examined the influence of a vegetarian or vegan diet on endurance performance. Five of these studies chose a cross-sectional study design, which means that without exercise intervention, the baseline performance levels of vegans/vegetarians were generally compared with omnivores.

Król and colleagues [27] compared the absolute exercise capacity (VO_2_max) and peak power output (PPO) of vegan (*n* = 22) and omnivorous (*n* = 30) amateur runners who trained at least three times a week. Weekly training volume did not differ between groups. Exercise capacity was assessed on a treadmill as absolute (L/min) and relative VO_2_max (mL/kg bodyweight/min) as well as PPO in watts. While no difference in absolute VO_2_max was detected between groups, relative VO_2_max was significantly higher in vegans compared to omnivores (*p* < 0.05) due to the significantly lower body weight (*p* < 0.05) of vegans compared to omnivores. PPO showed no difference between groups. Overall, no difference in oxidative capacity was detected between modes.

Lynch and co-authors [59] compared the cardiorespiratory fitness in a mixed cohort of 27 vegetarian and 43 omnivorous elite runners. Cardiorespiratory fitness was determined according to the Bruce protocol and expressed as VO_2_max [71].

The results revealed a significantly higher relative maximal oxygen uptake in the vegetarian diet group compared to the omnivorous group in females (*p* < 0.05) but not in males. In contrast, the absolute maximal oxygen uptake (L/min) did not differ between groups. The higher relative maximal oxygen uptake of the vegetarian females in this study was also explained with a lower body mass in vegetarians compared to the omnivores (0.05 < *p* < 0.10).

Another study yielded similar results [60]. This cross-sectional study of 28 strict vegetarians (can be stated as vegans) and 28 omnivores, who performed aerobic exercise for 196.3 min/week and 196.8 min/week, respectively, showed that vegans had both a higher estimated VO_2_max (44.5 ± 5.2 vs. 41.6 ± mL/kg bodyweight/min; *p* = 0.03) and higher submaximal endurance time to exhaustion (12.2 ± 5.7 vs. 8.8 ± 3.0 min; *p* = 0.007) than omnivores on a cycle ergometer. VO_2_max was calculated using a validated prediction equation adjusted for body weight: VO_2_max (mL/kg/min) = 10.8× (power output [W]/body weight [kg]) + 7. W is the maximal power output the participants achieved during the incremental cycle ergometer test. Body weight did not differ between groups (*p* = 0.8).

In contrast, Nebl and colleagues [70] did not observe differences in exercise capacity between lacto-ovo vegetarian, vegan and omnivorous recreational runners with similar training habits with a tendency of higher running distance (*p* = 0.054) and running time per week (*p* = 0.079) for lacto-ovo vegetarians. As a primary outcome, maximal power output (P_max_/bodyweight) that was reached in a graded exercise test on a bicycle ergometer was measured. As a secondary outcome, maximum power output related to lean body mass (P_max_/LBM) was assessed. There was no statistical difference in BMI (*p* = 0.559) and LBM between groups (*p* = 0.866). This indicates that all examined diets did not affect exercise capacity between groups.

A recent study [61] compared the cardiovascular fitness of nine habitual vegan and 16 habitual omnivorous young, healthy men by assessing the relative and absolute VO_2_max on a cycle ergometer. The data indicated no differences between groups for both relative and absolute VO_2_max.

These results suggest that long-term vegetarian and vegan diets do not have a detrimental effect on endurance performance, but may have the potential to improve endurance performance when performing exercise intensities relying on higher carbohydrate usage.

Four studies used an experimental approach. Blancquaert and colleagues [62] assigned 40 healthy female omnivores to either an omnivorous group (*n* = 10), a vegetarian group that was supplemented with creatine and β-alanine (*n* = 15) or a vegetarian group that received a placebo (*n* = 15) over a period of six months. Groups did not differ in age, height, weight and BMI. At baseline, 3 months and 6 months, subjects performed an incremental cycling test to assess VO_2_max (mL/kg body weight). VO_2_max did not differ between groups at baseline, nor did it change during the 6-month intervention period. Energy and macronutrient intake did not differ between the omnivorous group and the vegetarian groups.

In another study [44], 20 healthy, physically active (2–3 h of sport weekly) omnivores were allocated to either a lacto-ovo vegetarian or a mixed diet group and the influence of diet and a five-week sprint training program on power output, measured as 6×6 s repeated sprint ability on a cycle ergometer, was examined. To avoid a creatine deficiency, both groups supplemented 1g creatine monohydrate daily. Five weeks of sprint training led to an increase in power output per sprint (*p* < 0.05) and an increased mean power output for all sprints together (*p* < 0.001) independent of diet group (*p* = 0.707). No differences in energy and macronutrient intake were reported. These data do not show any superiority of either diet in terms of endurance performance or trainability.

Hietavala and co-authors [63] conducted a cross-over design study with nine healthy recreationally active men. Subjects were assigned to both a low-protein vegetarian and an omnivorous diet for four days each, separated by a 10–16-day washout phase. The data showed a significantly higher energy (*p* < 0.05), protein (*p* < 0.001) and fat intake (*p* < 0.01) in the omnivorous diet compared to the low-protein vegetarian diet. After the low-protein vegetarian diet, VO_2_ was significantly higher at 40% (*p* = 0.035), 60 (*p* < 0.001) and 80% (*p* < 0.001) of VO_2_max compared to the omnivorous diet, but no differences in exercise time to exhaustion between diets were detected. In fact, as no differences in RQ, plasma free fatty acids or triglycerides, plasma lactate or glucose contents were detected between groups, a changed use of substrates in energy production seems not to be an explanation of the differences in oxygen consumption. Further research is needed to elucidate this topic.

Another study was carried out on patients with type 2 diabetes [64]. In this study, 37 participants were respectively assigned to a hypocaloric (−500 kcal) vegetarian or hypocaloric conventional (omnivorous) diet group. Both groups performed aerobic exercise three times a week for 12 weeks. Two sessions were performed at 60% of maximal heart rate for 1h under professional supervision at the sports center, and the third session took place either at home or at the sports center. The results revealed a significant 21% increase in maximal performance (P_max_) (*p* < 0.001) and an increase in VO_2_max by 12% (*p* < 0.001) in the vegetarian diet group, but no significant changes in the omnivorous diet group, indicating that a vegetarian diet leads more effectively to an improvement in physical fitness in type 2 diabetes patients than an omnivorous diet.

Summarized, these studies do not unequivocally anticipate a superior role of vegan or vegetarian diet concerning performance, but detected a tendency of improved aerobic performance, which leads to the question of whether or how nutrition influences trainability and molecular adaptations.

## 5. Vegan and Vegetarian Diets and Strength Performance

### 5.1. Properties of Strength Performance

Muscular strength is the ability to generate force by skeletal muscle [72]. This essential physiological mechanism depends on several factors [72].

A main factor for muscular strength is the availability of phosphagens [73]. As strength performance is of shorter duration compared to endurance performance, but usually carried out with a higher power output, phosphagens such ATP and creatine phosphate are the predominant substrates for energy provision during resistance exercise [74,75]. It was shown that type II muscle fibers had a higher creatine content than type I muscle fibers (*p* < 0.01), and the ATP content of both fiber types, but especially that of type II fibers, were greatly reduced after a 25s maximal isokinetic cycling ergometer exercise (*p* < 0.01) [76]. Because of the lower dietary creatine intake, both blood and muscle creatine concentrations are lower by about 50% in plasma, by 35–39% in serum, and by 27–50% in red blood cells, in vegetarians compared to omnivores [13]. Creatine values of the less-restricted vegetarian diets were shown to be located between omnivorous and vegan values [3,77].

Therefore, fiber type distribution and the rate of anaerobic supply of ATP is critical to strength performance [75].

Second are the properties of the contractile elements. The distribution of slow-twitch type I muscle fibers and fast-twitch type II muscle fibers [78] varies between athletes according to the demands in their particular sporting discipline. Sprint runners have a lower percentage of slow-twitch fibers compared to distance runners [79] and strength trained individuals have a higher cross-sectional area (CSA) of type II muscle fibers compared to sedentary and endurance-trained individuals [80]. It has been shown that a caloric restriction of diet in rats led to decreased muscle weight and fiber area but did not affect neither the muscle fiber composition nor the muscle fiber transformation from type I to type IIA or IIB and vice versa [81]. To our knowledge, no studies on diet and muscle fiber type transformation in humans exist.

Third is the neuromuscular system. Research showed that four weeks of eccentric training led to adaptions of the central nervous system, resulting in an increased EMG activity of the agonist muscles during isometric activity and a decrease in the antagonists coactivation in concentric and eccentric actions (*p* < 0.05) [82]. However, it was also shown that a leucine-enriched protein supplementation did not influence neuromuscular adaptations in older adults [83], suggesting that the differences in protein intake between vegans, vegetarians and omnivores [3] do not affect neuromuscular adaptations to strength training. Vegan diets contain fewer amounts of leucine [65,84] and the role of leucine in skeletal muscle adaptation to strength training will be described in the following chapter. However, further research on this topic is needed.

Summarized, strength performance depends highly on substrate availability, the properties of contractile elements and the neuromuscular system. Nutritional differences between diets may affect phosphagen levels and muscle mass and thereby have an impact on strength performance.

### 5.2. Nutritional Aspects and Strength Performance

Strength performance is considerably affected by the nutritional behavior of the athlete [85,86]. After resistance exercise with 70% 1RM, muscle protein synthesis (MPS) increases up to four-fold compared to baseline [87]. In the fasted state, both MPS and muscle protein breakdown (MPB) increase after resistance training, while maintaining a negative muscle protein balance [88]. Therefore, nutrition in combination with resistance exercise promotes muscle anabolism [88]. Ingestion of dietary protein, particularly essential amino acids (EAA), after resistance training augments MPS and attenuates the exercise-induced increase in MPB, leading to a positive muscle protein balance [88]. The persistence of an EAA deficit throughout training would therefore lead to a maladaptation, as muscles cannot be remodeled without amino acids. It has been shown that both magnitude and duration of MPS can be enhanced if dietary EAA availability is increased after exercise [89]. Research showed that the consumption of both a low-dose (6.3 g) and a high-dose (12.6 g) essential amino acid (EAA) beverage led to a reduced protein breakdown compared to the consumption of 12.6 g whey protein [90]. Furthermore, both whole body protein synthesis and muscle protein synthesis were greater after ingestion of the EAA beverages compared to whey protein. As the administration of EAAs and mixed amino acids (MAA) resulted in a similar net protein balance after resistance training, non-essential amino acids do not appear necessary to elicit an anabolic response from muscle [91]. These results suggest a crucial impact of EAAs on muscle net protein balance. Additional ingestion of leucine with a meal-like amount of protein resulted in a greater MPS and a higher dietary protein incorporation into muscle protein [92]. Underpinning the importance of leucine for MPS, data showed that a low-protein (6.25 g) mixed macronutrient beverage can increase MPS as effectively as a high-protein beverage (25 g) if supplemented with additional 5.0g of leucine [93]. The crucial role of leucine in adaptations to strength training is discussed in the section on how vegan, vegetarian and omnivorous diets nutrition may affect molecular regulators of exercise adaptation.

Research on how regular dietary patterns affect MPS is sparse, but it has been shown that MPS increased both in young and elderly subjects by about 51% after ingesting a 113.4 g lean ground-beef patty [94]. In a more recent study, it was shown that MPS increased by 108% during the 5h period following a meal (340 g serving: 660 kcal, 90 g protein, 33 g fat) and a bout of resistance exercise. The more than doubled MPS can be attributed to both the higher protein intake and the addition of resistance exercise. These results show that protein-rich meals can increase MPS [95]. It should be noted that protein quality and quantity play a crucial role in stimulating MPS [96]. It is widely accepted that animal-derived proteins are higher in quality compared to proteins from plant sources [97,98]. Post-prandial muscle protein synthesis responses after ingestion of animal-derived proteins are higher compared to the ingestion of an equivalent amount of plant-based proteins [99]. The amino acid profile of plant-derived protein can be improved by combining plant sources [84]. MPS can also be enhanced by consuming a greater amount of plant protein [100].

In summary, muscle mass and strength performance depend on a positive muscle protein balance over extended time courses. This can be achieved by adequate protein and essential amino acid intake in combination with resistance exercise.

### 5.3. Vitamin D and Strength Performance

Vitamin D can be obtained either from diet or from sun exposure [101].

Within cultured chick myoblasts, it has been shown that vitamin D receptors (VDR) translocate from the nucleus to the cytoplasm rapidly (1–10 min) after exposure to the biologically active form 1,25(OH)_2_D_3_ [102]. When binding to the VDR in the cytoplasm, 1,25(OH)_2_D_3_ elicits rapid uptake of calcium within the muscle cell, implying a non-genomic role for calcium handling and muscle function [103].

Research on the relationship of vitamin D levels and muscle function generate ambiguous results.

Cross-sectional studies display correlations of vitamin D levels and muscle function.

Lower 25(OH)D serum concentrations were correlated with lower knee extension strength (*r* = 0.08, *p* = 0.020) and flexion strength (*r* = 0.07, *p* = 0.032) in 75-year-old women [104].

In contrast, another study detected no consistent association between serum 25(OH)D and muscle mass (total body dual-energy X-ray absorptiometry) or muscle strength (handgrip force and isometric knee extension moment) in 311 men (22–93 years old) and 356 women (21–97 years old) [105].

However, there was a significant association between low 1,25(OH)_2_D_3_ levels and low skeletal mass in both men (*p* = 0.041) and women (*p* = 0.001) and low isometric knee extension moment (*p* = 0.018) as well as handgrip force (*p* = 0.026) in women when subjects were younger than 65 years [105]. The association between low vitamin D levels and low skeletal muscle strength needs further research as findings are inconsistent [106,107].

Summarized, the substrate availability, the properties of the contractile elements, neuromuscular adaptions, protein (especially essential amino acid) and vitamin D intake can influence strength performance. As already mentioned, diet is partly capable of altering these factors and therefore, due to the restrictive dietary pattern, vegetarian and vegan diets may impact strength performance differently than omnivorous diets.

### 5.4. Evidences on Vegetarian and Vegan Diets and Strength Performance

Eight studies examined the influence of a vegetarian or vegan diet on strength performance. Three of these studies chose a cross-sectional study design. Lynch and colleagues compared isokinetic leg extension strength of 27 vegetarian and 43 omnivore elite runners at angle velocities of 60°/s, 180°/s and 240°/s [59]. The results showed no difference of peak torque when performing leg extension, suggesting that a vegetarian diet may adequately support strength.

Another study compared 28 vegan and 28 omnivorous lean physically active women. Muscle strength was assessed using a leg press and a chest press machine and measured using the one repetition maximum (1RM). Additionally, muscular strength indices were calculated for both the leg press and the chest press and expressed as weight lifted in kg per kg lean body mass [60]. Lean body mass in subjects was not significantly different (*p* = 0.8). The results showed a tendency for decreased upper body muscle strength in vegans (*p* = 0.06) but no differences in lower body muscle strength (*p* = 0.5).

A recent study compared the lower body strength of 16 habitual omnivorous and nine habitual vegan healthy, young men [61]. Therefore, subjects performed knee extension maximal voluntary isometric contraction on an isokinetic dynamometer. The data showed no differences between groups.

Based on these results, the authors conclude that a vegan diet seems not to have a detrimental effect on muscle strength in healthy young, physically active individuals. This suggests that a vegan diet may be adequately supportive to maintain muscle strength.

The remaining five studies used an experimental approach. In one study [66], 21 male subjects were allocated to a self-selected lacto-ovo-vegetarian (LOV) diet for two weeks to familiarize with the dietary pattern. After these two weeks, baseline strength assessment was conducted for five exercises (knee extension, seated leg curl, double leg press, seated arm pulldown, seated chest press). After baseline measurements, the participants were randomly divided into two dietary groups. One group received 0.6 g protein/kg bodyweight/day of beef products additionally to their LOV diet, the other group received 0.6 g protein/kg bodyweight/day texturized vegetable protein meat-analog products (TVP). Over the following 12 weeks, subjects participated in resistance training on three nonconsecutive days per week at 80% of their assessed 1RM. Strength assessment was conducted at baseline, after five weeks and after 12 weeks of resistance training. Baseline strength values showed no significant differences between groups. Maximal strength (1RM) increased significantly (*p* < 0.05) in all of the trained muscle groups by 14% to 38%. No difference between the TVP and the meat group were detected in 4/5 exercises. The TVP group had a greater increase in strength for the knee extension exercise (group × time interaction [*p* < 0.01]) compared to the beef group. Body weight, energy and macronutrient intake did not differ between groups at baseline, 5 weeks and 12 weeks of intervention. These results suggest that a vegetarian diet may not have a detrimental effect on muscular strength, but on the contrary tends to be more beneficial to strength performance than the beef-containing diet, as indicated by the increased strength for the knee extension exercise.

Haub and colleagues [67] used a similar study design, as participants underwent a two-week baseline period, during which they familiarized with an LOV diet and TVP, followed by a 12 week intervention with resistance training and protein intake standardization [66]. The resistance training sessions consisted of two sets of eight repetitions at 80% 1RM and a third set until voluntary fatigue. Upper body (Newton per second) and lower body power output (Newton meter per second) were assessed at 20%, 40%, 60% and 80% of the previously tested 1RM. After 12 weeks of resistance training, power output was retested. The results showed an increase in lower body and upper body 1RM after the 12-week resistance training program. No differences between groups concerning muscular strength and power output were detected. Energy and macronutrient intake did not differ between groups at baseline and post-intervention. These results indicate that both diets are equally effective at improving muscle strength and power output.

In a previous study by Haub and colleagues [68], participants underwent a study protocol similar to the one aforementioned [66]. Body weight, fat-free mass and fat mass were not significantly different between groups before the intervention and remained unchanged throughout the study. Energy and protein intake were not significantly different either between groups. Muscle strength (1RM) increased significantly (*p* < 0.05) for all muscle groups trained (unilateral seated leg extension, unilateral seated leg flexion, bilateral leg press, seated chest press, arm pull) independent of diet. These results suggest that the predominant source of dietary protein does not influence the increase in muscle strength.

The results of another study [69] support this notion. In this study, overweight participants (*n* = 19) were allocated to two groups. One group maintained their habitual (omnivorous) diet (*n* = 9), the other group was counseled to self-select a LOV (*n* = 10) diet. After assessing baseline measurements, subjects participated in a 12-week resistance training program with two nonconsecutive sessions a week, performing two sets of eight repetitions at 80% 1RM and a third set until muscular fatigue. Tests and evaluations were carried out at baseline, week 6 and week 12 of RT. The results showed a significant increase in dynamic muscular strength in the exercised muscle groups in both dietary groups. No significant differences in baseline strength and strength increases throughout the 12-week period of resistance training between dietary groups were detected, but fat-free mass and skeletal muscle mass increased in subjects with a meat-containing diet and decreased in subjects with a lacto-ovo-vegetarian diet.

In a recent study [65], a group of habitual vegans (*n* = 19) and a group of habitual omnivores (*n* = 19) performed strength training twice a week over a period of 12 weeks. Habitual protein intake was assessed at baseline and adjusted to 1.6 g/kg bodyweight/day via supplemental protein. After the intervention, leg lean mass, rectus femoris CSA, vastus lateralis CSA, vastus lateralis muscle fiber type I and type II CSA and leg-press 1RM increased significantly compared to baseline with no differences between groups.

These findings lead to the conclusion that a vegetarian and vegan diet can be sufficient for strength improvement, but are inferior to a meat-containing diet regarding an increase in fat-free mass and skeletal muscle mass. These findings lead to the question of whether or how diet influences trainability and molecular adaptions and as a consequence strength performance.

## 6. Vegan, Vegetarian and Omnivorous Diets May Affect Molecular Regulators of Exercise Adaptation in Human Skeletal Muscle

Regardless of the type of diet, any adaptation of skeletal muscle towards exercise depends on the coordinated activation of molecular signaling pathways [108] (Figure 2). Resistance exercise (RE) as well as endurance exercise (EE) adaptations in skeletal muscle are regulated by diverse subsets of molecular pathways that ensure a very specific adaptation in muscle.

While RE increases the synthesis of mainly sarcoplasmic and myofibrillar proteins and increases strength, EE induces increased mitochondrial protein synthesis [109], the formation of new capillaries [110], and enhances cardiac adaptations [22]. The entire subsets of proteins, metabolites and transcriptomic responses that are differently occurring between RE and EE are still being unraveled [111,112,113,114]. Therefore, it is not yet precisely described whether vegan or vegetarian diets may enhance or even blunt the molecular exercise adaptation towards RE or EE compared to omnivorous diets. This process could occur mainly on two levels.

Firstly, the composition of the diet may have a direct effect on the acute molecular adaptation process in skeletal muscle after exercise. Secondly, the diet may modulate the gut microbiome, which then indirectly but permanently changes the systemic environment in the organism [115] to modulate skeletal muscle adaptation [116] and nutrient uptake [117].

### 6.1. Proteins and Amino Acids and Their Impact on Molecular Signaling

The main macronutrients that significantly drive the adaptation towards RE are proteins and their molecular building blocks amino acids [118]. Resistance exercise drives the mechanically-induced activation of mTORC-1 signaling initiating ribosome activity and protein synthesis [119] which depends also on the availability of amino acids in skeletal muscle [120]. Leucine is an essential amino acid that activates mTOR signaling [121] after entrance into the muscle cell via LAT1 amino acid transporters [122]. Protein administration rapidly elevates amino acid levels in the blood stream, increases the abundance of amino acid transporters [123] and consequently the uptake of amino acids within muscle. Therefore, protein administration is generally accepted to increase muscle protein synthesis significantly above the levels of RE when carried out in the fasted state [124]. This emphasizes also a critical role for protein uptake in combination with RE in the elderly [125].

A reduction in caloric intake and especially proteins [126] may reduce muscle mass rapidly in the elderly [127] and in younger individuals [128,129], while increased protein levels may preserve muscle mass [129]. Nutritional imbalances, especially due to reduced protein intake can be found in aging adults. This has been shown to be involved in delineating the nutritional frailty phenotype in the elderly [130,131]. Data show that an increase in protein intake in aging subjects is an important aspect to maintain and increase muscle mass and moreover to substantially enhance muscle function [132]. However, whether a vegetarian or vegan diet poses a risk of insufficient protein provision for the elderly is discussed controversially [133].

Meanwhile, the growing scientific knowledge concerning mechanisms and consequences of protein supplementation formed a stable basis for athletes [128,134], but also for the fitness-associated and aging population [135]. However, it is still investigated and discussed [84] whether differences in protein composition between vegan, vegetarian and omnivorous diets may differently modulate adaptability and performance towards RE [65,136]. Ciuris and colleagues determined [98] that subjects consuming vegan protein sources may require an additional 10 g of protein per day as digestibility and metabolism are less efficient in vegan protein sources. Additionally, distinct protein preparations, e.g., soy, egg-, milk- and beef-derived proteins can differ in the kinetics and rate of amino acid uptake in the body [137]. Long term effects of RE under application of soy vs casein exerted unequivocal results concerning strength and increases in muscle mass. Some studies observed no difference between plant-based soy protein vs milk protein concerning performance [138], while others showed augmented effects of milk compared to soy protein [139,140].

### 6.2. Creatine and Its Impact on Molecular Signaling

The mTOR complex is regulated on many levels including mechanical stimulation [119], amino acid abundance [120] and also by growth factors such as insulin-like growth factor-1 (IGF-1) [141]. It has been shown that the mRNA levels of IGF-1 can be increased by the supplementation of creatine in cultured myotubes [142] and in human skeletal muscle [143] and is associated with muscle hypertrophy [142]. Therefore, It may be hypothesized that creatine consumption due to OMV, VGT and VEG nutritional habits may differently affect skeletal muscle adaptation. However, although IGF-1 accumulation in skeletal muscle fibers is indeed increased upon supplementation with creatine, VGT-related subjects showed similar responses to creatine supplementation compared to OMV-related subjects [144].

### 6.3. Vitamin D and Its Impact on Molecular Signaling

It has also been shown that vitamin D affects molecular processes that can influence muscle strength [103]. The biologically active form 1,25(OH)_2_D_3_ binds to the specific VDR that is located both in the cytoplasm and the nucleus [103]. VDR expression is strongly upregulated following injury [145] and the overexpression of VDR in rat skeletal muscle leads to increases in anabolic signaling, ribosomal biogenesis and protein synthesis, resulting in increased skeletal muscle hypertrophy [146]. Evidence suggests that genomic responses to 1,25(OH)_2_D_3_ down-regulate myoblast proliferation and enhance differentiation into myotubes, as shown in cultured rat and mice myoblast cells [147].

The habitual vegan diet is lower in dietary protein and vitamin D than the habitual omnivorous diet [3,65]. Plant-based diets have also been shown to contain significantly fewer amounts of essential amino acids in general, and leucine in particular (*p* < 0.05) [65,84]. These differences in amino acid composition between plant-based and animal-based proteins are thought to be related to the inferior postprandial MPS of subjects consuming a habitual vegan diet [84], as it has been shown that beef stimulated postprandial MPS to a greater extent than an isonitrogenous amount of a soy-based beef replacement [134].

Still, the current guidelines for protein administration range from 1.2 g to 2.2 g per kg of bodyweight [1,86].

However, it was recently proposed that vegan protein sources will have a similar effect on amino acid uptake and muscle anabolism, when a higher amount of vegan protein is consumed, different sources of protein are ingested or additional doses of leucine are consumed [84,136].

Summarized, dietary regimens are capable of affecting MPS by both providing proteins as building blocks for muscle tissue and activation of molecular signaling pathways through leucine, vitamin D and creatine. Since vegetarians and vegans consume less of these nutrients compared to omnivores, a vegetarian or vegan diet might affect muscular adaptation negatively. The regular composition of omnivorous diets more strongly supports the adaptive potential towards resistance exercise.

### 6.4. Polyunsaturated Fatty Acids May Augment Skeletal Muscle Adaptation in Response to Exercise

With the exception of pesco-vegetarians, vegetarians and vegans consume a significantly lower amount of polyunsaturated fatty acids (PUFAs) [3,148]. Fatty acids are used as substrates for oxidative metabolism [149] and stored in lipid droplets within skeletal muscle fibers adjacent to mitochondria [150]. Generally, fatty acids and related compounds have important roles for composing the architecture and rebuilding of cell membranes during tissue turnover [151,152]. The chemistry, metabolism and fate of fatty acids in the physiological environment and their role for exercise performance is complex [153]. In some animals, the administration of omega-3 fatty acids EPA and DHA via natural food intake has received great attention. Birds like the sandpiper (calidris pusilla) that perform long-haul flights once a year, travel enormous distances while the energetic demand must nearly exclusively depend on fatty acid oxidation [154]. The energetic environment in flight muscles requires a sufficient number of mitochondria and a dominantly fatty-acid-based fuel to spare weight (glycogen has a much lower energy density and weighs more), while also preserving enough energy for several days of non-stop repetitive muscle contraction [154]. Before they start their journey, they consume increased amounts of small crabs (corophium volutator) containing a significant amount of PUFA fatty acids, especially EPA and DHA [155]. Although this kind of diet is far from being vegan or vegetarian, the oxidative environment of skeletal muscle is significantly increased during this time. This is regulated by the persistent activation of molecular signaling towards significantly enhanced mitochondrial adaptation [156], importantly: without increased training. Moreover, in these birds, increased transport capacity of fatty acids towards the mitochondria is substantially enhanced due to an increased membrane permeability mediated by the incorporation of DHA and EPA in cell membranes [157]. PPAR receptors (PPAR alpha, beta and gamma) are substantially activated by these fatty acids and mediate the communication between fatty acid availability and adaptation [158]. Indeed, these receptors serve as molecular switches that sense and bind these fatty acids to link nutrition-dependent signals to a PPAR-dependent signaling. This affects the transcription of proteins involved in fatty acid metabolism as well as augmented PGC1-alpha signaling to mediate mitochondrial adaptation [159,160].

While beneficial effects of DHA and EPA on migrating birds are an extreme example for the physiological relevance of this mechanism, studies have also determined detrimental effects of those fatty acids. It was determined that cell proliferation in vitro [161] as well as myogenesis and mitochondrial biogenesis in developing mice [162] were reduced. Additionally, there seem to be substantial dose-dependent effects, as high doses were shown to switch myogenesis to adipogenesis in C2C12 primary muscle cells [163]. Therefore, to which extent fatty acid-mediated signaling may drive enhanced muscle adaptation in humans is still investigated. Findings from animal studies determined increased satellite cell proliferation upon EPA and DHA treatment [164,165], while evidence for enhanced satellite cell proliferation in humans and thus the potential to support the growth potential of skeletal muscle in the long-term has not been shown so far [166]. However, some human studies show that muscle mass and strength can be augmented under EPA and DHA administration [167] and phosphorylation of mTOR-related signaling as well as protein synthesis can be increased [168], while others observed no increase after acute RE compared to placebo treatment [169].

In summary, beneficial effects for strength and muscle mass in humans are partly inconclusive [170] and the overall effect of those fatty acids on EE performance is considerably lower in humans than in migrating birds [171,172,173].

Nevertheless, given that omnivorous, vegetarian and vegan diets do not differ in terms of total fat intake, but they do differ significantly in DHA intake (182 mg, 33.8 mg, 18.2 mg, respectively, standardized to 2000 kcal/d) [3], it may trigger exercise-induced adaptation towards EE and RE in a more subtle but persistent manner and importantly more in omnivores. The pesco-vegetarian diet is an exception that contains high levels of DHA and EPA due to the consumption of fish [3,148]. DHA and EPA are not considered essential since they can be converted from alpha-linolenic acid (ALA) (at a conversion rate of about 5–8%) [148]. As plant-based foods containing ALA are also high in linoleic acid (LA), the nutritional challenge for vegans and vegetarians is to increase dietary ALA without increasing dietary LA, because these fatty acids compete for the same biochemical pathway for conversion to EPA and arachidonic acids (AA), respectively [148].

Besides PUFAs, the amino acid taurine is also capable of altering muscular molecular signaling [174,175]. The knockout of taurine transporters in mice led to reduced levels of PPARα and its transcriptional targets [174], whereas taurine supplementation increased the activation of AMP-activated protein kinase (AMPK) in mice [175]. AMPK is a major energy sensor in skeletal muscle that regulates energy homeostasis [176] and mitochondrial biogenesis [177] by increasing the phosphorylation and expression of PGC1-alpha [178]. As taurine is highly abundant in beef and absent from plants [43], dietary choices may possibly affect molecular adaptation and performance.

In summary, polyunsaturated fatty acids like EPA and DHA as well as the amino acid taurine provide a significant molecular potential to enhance skeletal muscle adaptation due to an increased nutritional uptake. However, despite findings from in vitro studies, there is no clear evidence that either an increased natural uptake based on the choice of diet or an increased artificial uptake of EPA/DPA via supplementation significantly increases tissue adaptation and exercise performance in humans.

## 7. Influence of Diet on the Microbiome and Its Effect on Exercise Performance and Basal Molecular Signaling

The gut microbiome may have a collective genome size 150-fold that of the human, and it has been argued that because of its metabolic capacity, it merits the consideration as an organ in its own right [179]. The microbiota of a healthy individual is diverse and the majority of the microbial communities are symbiotic and commensal [180]. It has been shown that the microbiota can be modulated by exercise training [180,181] and diet [180,182]. Modulations caused by exercise affect the epithelial cells integrity and intestinal epithelium permeability [116]. High volume endurance training increases epithelium permeability, promoting the passing of bacterial toxins and pathogens to pass into the bloodstream [183]. As a consequence, NF-κB-dependent inflammatory pathways as well as FOXO-dependent muscle degradation pathways are activated and adaption to exercise is negatively affected [116]. In vitro experiments showed that FOXO promotes atrophy of muscle mass in mice [184,185,186]. Experiments in rodents showed that the activation of NF-κB caused atrophy in skeletal muscle whereas the inhibition of this pathway prevented atrophy [187,188].

Changes of the gut microbiome through diet already occur after 24 h and will reverse to baseline 48 h after discontinuation [182]. These changes include carbohydrate and protein fermentation processes [189,190], intestinal inflammation [190], fat oxidation [191] and might also be capable of promoting protein anabolism by increasing amino acid availability [116,181]. Modulation of the immune response, oxidative stress, metabolic processes, and nutrient bioavailability are considered as the main mechanisms by which the microbiota affects training adaptation [116]. Intestinal microbiota may contribute to myocyte anabolism by alleviating farnesoid X receptor (FXR) that plays an important role in metabolic pathways, lipoprotein and glucose turnover [116]. Another contribution of the gut microbiome for improving human body physiology is the synthesis of short-chained fatty acids (SCFA), the end products of fermentation of dietary fibers in the intestines [192]. The level and ratio of the different SCFAs (molar ratio of 60:20:20 in acetate, proprionate and butyrate) [193] are key parameters for microbiota and mucosa health [180]. Providing about 10% of the daily caloric requirement [192], SCFAs can be used as energy-deriving substrate for numerous tissues including muscle, indicating that they can contribute to enhanced skeletal muscle growth [194]. SCFA produced by intestinal bacteria have a positive effect on the integrity of the intestinal barrier, protecting it against inflammation [116]. Furthermore, SCFAs are discussed as putative signaling molecules for skeletal muscle adaptation of skeletal muscle [192]. SCFA can directly phosphorylate and activate AMPK by increasing the AMP/ATP ratio in skeletal muscle [116,192].

As vegan diets contain significantly more fiber than in OMV and VGT [3], a higher basal SCFA synthesis may therefore increase the basal molecular activation of AMPK and PGC1-alpha, a molecular basis for increased capacity for oxidative metabolism and fatty acid oxidation in VEG [178].

Excessive protein intake causes an increase in the number of protein fermenting bacteria and decrease of number of carbohydrate-fermenting bacteria. By-products of fermentation of undigested protein such as ammonia, biogenic amines, indole compounds, and phenols are mainly toxic and may exacerbate the inflammatory response and increase tissue permeability, therefore being detrimental to gut health [116,195]. Diets with a high protein content would increase small intestinal pH, favoring the proliferation of pathogenic bacteria. When switching to a diet with reduced protein intake, microbial composition shifts towards higher counts of beneficial, carbohydrate-fermenting bacteria [195]. In addition to the quantity, the quality of dietary protein may also influence protein fermentation within the gastrointestinal tract. Highly digestible proteins, like casein, can be digested in the proximal intestine, resulting in less undigested proteins for fermentation in the distal intestine [195]. Plant-derived protein are not completely digested in the proximal intestine, resulting in microbial fermentation in the distal intestine. As a result, the source of protein, and as a consequence residual protein volume, affects the composition of bacterial groups involved in protein fermentation [195]. By consuming dietary protein with a high digestibility, the amount of dietary protein reaching the distal intestine can be diminished, leading to a suppression of the growth and activity of potential pathogens. Studies on the effect of dietary protein on gut microbiome composition is ambivalent and needs further research (for review see [195]).

Excessive fat intake may also significantly affect the composition of the intestinal microbiota, limiting substrates for SCFAs production [116].

High-fat diet also reduces the diversity of bacterial strains and the abundance of Bacteroidetes, which are considered the leading factor of gut homeostasis and health while promoting the growth of Firmicutes and Proteobacteria [116,196], the latter having inflammatory properties [197]. Research shows ambivalent results about the effect of diet on Firmicute abundance. Research of Hills and colleagues report a higher ratio of Firmicutes to Bacteroidetes in the gut of omnivores and in obese subjects compared to lean subjects [197], whereas Jandhyala and colleagues report a decrease in Firmicutes as a result of an omnivorous diet [198].

Vitamin D plays an essential role in maintaining a healthy gut microenvironment [180]. Considering the variety of functions of vitamin D, an inadequate level may impair intestinal homeostasis, since vitamin D can influence bacterial colonization and exert anti-inflammatory responses through interaction with VDR. VDR expression and location may be also regulated by commensal or pathogenic gut microbiota [180]. Vitamin D also contributes to maintenance of the integrity of the epithelial barrier [180,199].

Yet, the connection between gut microbiome and physical performance is not completely understood [181] and needs to be investigated more closely.

Summarized, the gut microbiome is strongly dependent on nutrient intake. A high fiber intake promotes SCFA production which has a positive effect on gut microbiome composition as well as molecular adaptions through activating AMPK, suggesting a favorable effect of a diet high in fiber on adaptation to EE. Excessive intake of protein, especially proteins with a low digestibility, affects the gut microbiome negatively by lowering the intestinal pH, favoring the proliferation of pathogenic bacteria. High-fat diets also reduce the favorable diversity of bacterial strains of the gut microbiome. Vitamin D contributes to intestinal homeostasis since it is capable of influencing bacterial colonization and has anti-inflammatory properties through interaction with VDR. As vegans’, vegetarians’ and omnivorous’ intake differ in these nutrients, dietary regimens might have an impact on gut microbiome health.

## 8. Summary and Future Directions of Research

Research on the influence of a vegan or vegetarian diet on exercise performance is scarce. Exercise performance is dependent on multiple physiological subsystems. Those can be affected either directly, during exercise, through the uptake of specific nutrients but also indirectly, by nutrient-induced modulation of the molecular environment that promotes e.g., muscular adaptations. Endurance performance depends on skeletal muscle mitochondrial and capillary density, hemoglobin concentration, endothelial function, functional heart morphology and availability of carbohydrates. The macro- and micronutrient composition of vegan and vegetarian diets implies potentially advantageous properties for endurance performance compared to an omnivorous diet.

Strength performance depends on factors that can be influenced by diet e.g., creatine and protein availability which alter muscle protein synthesis. Therefore, when not controlled, the macro- and micronutrient composition of vegan and vegetarian diets may elicit potentially disadvantageous properties for strength performance.

Although the impact of a vegetarian or vegan diet on molecular muscular adaptation has yet not been thoroughly investigated, the existing literature indicates the influence of particularly important nutrients, like leucine, taurine, DHA, EPA and SCFA on molecular signaling in tissues and in the long-term different diet regimens may therefore affect exercise performance.

Besides that, the choice of diet also influences the gut microbiome. It is widely accepted that the constellation and variety of the gut microbiome significantly affects mechanisms like intestinal inflammation, production of SCFA, fat oxidation, carbohydrate and protein fermentation processes, and protein anabolism. Vegan and vegetarian diets possess potentially beneficial properties for the gut microbiome and might therefore influence those mechanisms which may affect in the long-term exercise performance.

However, scientific research yet failed to show a robust difference of physical performance between diets.

To unravel the detrimental and beneficial aspects of the dietary choice on exercise performance, future studies must carefully combine the analysis of molecular signaling networks in combination with physiological read-outs in extended time frames. It must be considered, that upon dietary changes a multitude of metabolic pathways may change within the organism. Therefore, the use of blood metabolomics may be an important tool to study diet-induced changes in the metabolism.

## Figures and Tables

**Figure 1 nutrients-13-03884-f001:**
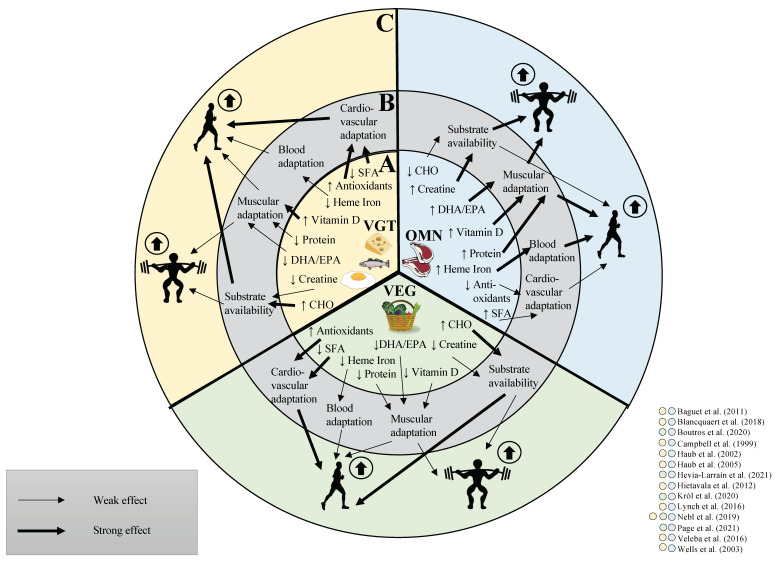
Impact of dietary properties on physiological subsystems and performance. (**A**): Vegan, vegetarian and omnivorous diets possess unique nutritional properties. This affects the intake of differential levels of polyunsaturated fatty acids (DHA/EPA), carbohydrates (CHO), creatine, protein, vitamin D, heme iron, antioxidants and saturated fatty acids (SFA). (**B**): The diet composition affects substrate storage and tissue adaptations on multiple levels and (**C**): finally can change strength and endurance performance in combination with physical exercise. The arrows describe a high occurrence (↑) and a low occurrence (↓) in the particular diet [27,44,59,60,61,62,63,64,65,66,67,68,69,70]. Omnivorous diets (OMN, blue section) possess higher amounts of DHA/EPA, vitamin D and protein which have a strong effect on muscular adaptation and therefore on strength and endurance performance. The high amount of creatine has a strong effect on substrate availability and therefore strength performance, whereas the low amount of CHO has a weak effect on substrate availability and therefore affect endurance performance to a lower extent. The high heme iron content has a strong effect on blood adaptation and therefore on endurance performance. Low antioxidant content and high amounts of SFA have a weak effect on cardiovascular adaptation, affecting endurance performance to a lower extent. Nevertheless, studies showed a significant increase in physical performance of an OMN diet when combined with physical exercise. Vegan diets (VEG, green section) possess low amounts of protein, DHA/EPA and vitamin D and therefore exert only a weak effect to support muscular adaptations for strength and endurance performance. The high amount of CHO has a strong effect on energy-deriving substrate availability and therefore endurance exercise. whereas the low amount of creatine has a weak effect on energy-deriving substrate availability and therefore on strength performance. Low heme iron levels have a weak effect on blood adaption therefore affecting blood adaptations to a smaller extent. Low amounts of SFA and high amounts of antioxidants have a strong effect on cardiovascular adaptations and therefore on endurance performance. Nevertheless, studies showed a significant increase in physical performance of a VEG diet when combined with physical exercise. Vegetarian diets (VGT, yellow section) possess low amounts of protein and DHA/EPA, and therefore have a weak effect on muscular adaptations and strength performance. In contrast, the higher amount of vitamin D has a strong effect on muscular adaptation and therefore on muscular adaptations. The high CHO content has a strong effect on substrate availability for endurance exercise, whereas the low creatine content has a weak effect on substrate availability for strength exercise. Low heme iron content has a weak effect on blood adaptation, therefore affecting endurance performance to a lower extent. High levels of antioxidants and low amounts of SFA have a strong effect on cardiovascular adaptations and therefore influences endurance performance to a greater extent. Nevertheless, studies showed a significant increase in physical performance of a VGT diet when combined with physical exercise.

**Figure 2 nutrients-13-03884-f002:**
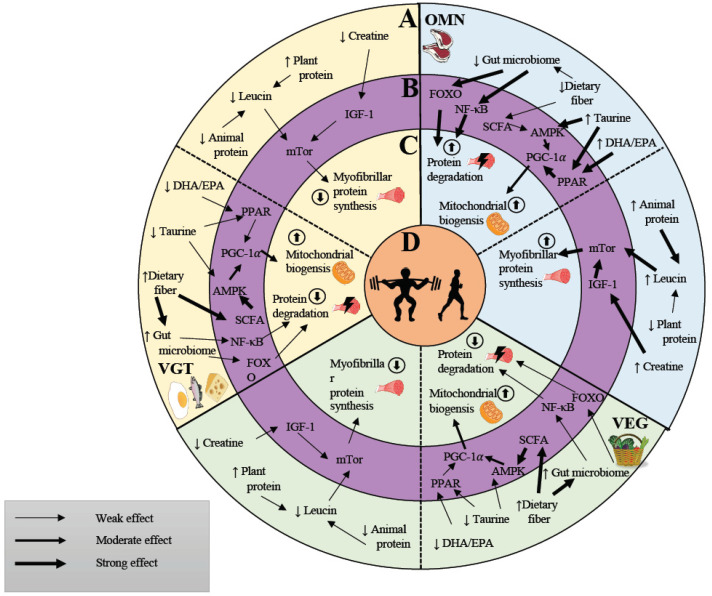
Influence of dietary properties on molecular signaling and muscular adaptation. (**A**): Vegan (VEG), vegetarian (VGT) and omnivorous (OMN) diets possess unique nutritional properties. This affects differential levels of polyunsaturated fatty acids, dietary fibers, plant- and animal-based protein sources, creatine and leucine. (**B**): Diet composition affects molecular signaling pathways. (**C**): Molecular signaling activates mitochondrial and myofibrillar protein synthesis and degradation and hereby modulates skeletal muscle adaptation and (**D**): exercise performance.Omnivorous diets (OMN, blue section) possess a lower amount of dietary fiber. This negatively affects the gut microbiome and reduces intestinal short chain fatty acid (SCFA) production. This induces increased FOXO and NF-κB signaling which can increase protein degradation. Reduced amounts of SCFA activate AMPK signaling to a lower extent which decreases AMPK-induced PGC-1α activation and affects mitochondrial biogenesis. In contrast, OMN diets contain elevated amounts of DHA/EPA and taurine, which enhances PPAR-induced PGC-1α activation. Taurine also activates AMPK signaling, leading to an overall moderate effect on mitochondrial biogenesis. OMN diets contain low amounts of plant-based protein sources but high amounts of animal-based protein with a higher leucine and creatine content. These two diet dependent factors lead to an activation of mTOR-based signaling which enhances the potential for increased myofibrillar protein synthesis (MFPS). Vegan diets (VEG, green section) possess a higher amount of dietary fiber. This positively affects the gut microbiome and enhances the intestinal SCFA production. This reduces FOXO and NF-κB signaling which leads to a decreased protein degradation. Increased amounts of SCFA activate AMPK to a higher extent which increases AMPK-induced PGC-1α activation and enhances mitochondrial biogenesis. In contrast, VEG diets contain reduced amounts of DHA/EPA and taurine which leads to a decreased PPAR-induced PGC-1α activation. The low taurine content also decreases AMPK activation leading to an overall moderate effect on mitochondrial biogenesis. VEG diets contain high amounts of plant-based protein but low amounts of creatine- and leucine-rich animal-based proteins. Therefore, a VEG diet result in a lower activation of mTOR-based signaling which reduces the potential for increased MFPS.Vegetarian diets (VGT, yellow section) possess a higher amount of dietary fiber. This positively affects the gut microbiome and enhances the intestinal SCFA production. This reduces FOXO and NF-κB signaling which leads to a decreased protein degradation. Increased amounts of SCFA activate AMPK to a higher extent which increases AMPK-induced PGC-1α activation and enhances mitochondrial biogenesis. In contrast, VGT diets contain reduced amounts of DHA/EPA and taurine which leads to a decreased PPAR-induced PGC-1α activation. The low taurine content also decreases AMPK activation leading to an overall moderate effect on mitochondrial biogenesis. VGT diets contain high amounts of plant-based protein but low amounts of creatine- and leucine-rich animal-based proteins. Therefore, a VEG diet result in a lower activation of mTOR-based signaling which reduces the potential for increased MFPS.

**Table 1 nutrients-13-03884-t001:** Properties of diets (adapted from [3,4]).

Type of Diet	Foods Included
Omnivorous	Eats red meat, poultry, fish, dairy and eggs
Semi vegetarian	Eats dairy, eggs and some red meat, poultry and fish ≥1 time/month but <1 time/week
Lacto-vegetarian	Eats dairy, but no red meat, poultry, fish or eggs
Ovo-vegetarian	Eats eggs but no red meat, poultry, fish or dairy
Pesco-vegetarian	Eats fish, but no red meat, poultry, dairy or eggs
Lacto-ovo-vegetarian	Eats dairy and eggs but no red meat, poultry or fish
Pesco-lacto-ovo-vegetarian	Eats fish, dairy and eggs but no red meat or poultry
Vegan	Eats only plant-based foods (no red meat, poultry, fish, dairy or eggs)

**Table 2 nutrients-13-03884-t002:** Macronutrient and micronutrient intake of diets (adapted from [3]). Mean nutrient intake values with standard errors (SE) standardized to 2000 kcal/day.

Nutrient	Omnivorous		Semi Vegetarian		Pesco-Vegetarian		Lacto-Ovo-Vegetarian		Vegan	
	Mean	SE	Mean	SE	Mean	SE	Mean	SE	Mean	SE
Caloric intake (kcal/day)	1890	4	1713	12	1937	9	1899	5	1894	10
Total carbohydrate (g)	266	0.2	283	0.7	284	0.5	286	0.3	309	0.6
Carbohydrate (% Energy)	53.1	<0.1	56.6	0.1	56.8	0.1	57.2	0.1	61.7	0.1
Total fiber (g)	30.4	<0.1	34.9 *	0.1	37.7 *	0.1	37.5 *	0.1	46.7 *	0.1
Total fat (g)	78.1	0.1	74.2	0.3	73.4	0.2	73.6	0.1	66.1	0.2
Fat (% Energy)	35.1	<0.1	33.4	0.1	33.0	0.1	33.1	0.1	29.8	0.1
MUFA (g) ^a^	32.4	0.1	30.5	0.2	30.9	0.1	30.3	0.1	28.0	0.1
SFA (g) ^b^	19.9	<0.1	17.4	0.1	15.8 *	0.1	16.0	0.1	11.6 *	0.1
DHA (g) ^c^	182	1.2	69.8 *	3.6	187	2.8	33.8 *	1.5	18.2 *	3
Total protein (g)	75.8	0.1	71.8	0.2	74.3	0.2	72.0	0.1	72.3	0.2
Protein (% Energy)	15.2	<0.1	14.4	<0.1	14.9	<0.1	14.4	0.1	14.5	<0.1
Animal protein (g)	31.8	0.1	17.6 *	0.2	16.0 *	0.2	12.2 *	0.1	3.1 *	0.2
Animal protein (% Energy)	6.4	<0.1	3.5 *	<0.1	3.2 *	<0.1	2.4 *	<0.1	0.6 *	<0.1
Plant protein (g)	43.9	0.1	54.1 *	0.2	58.2 *	0.2	59.7 *	0.1	69.2 *	0.2
Plant protein (% Energy)	8.8	<0.1	10.8 *	<0.1	11.6 *	<0.1	11.9 *	<0.1	13.8 *	<0.1
Vitamin D (μg)	10.6	0.1	9.9	0.2	9.8	0.2	8.6	0.1	6.3 *	0.2
Magnesium (mg)	509	1.3	554	3.7	581	2.9	567	1.6	652 *	3.1
Iron (mg)	32.9	0.3	34.1	0.9	34.6	0.7	34.1	0.4	31.6	0.8

* Significant contrast (*p* < 0.05 and a mean difference ≥20% when compared to omnivorous dietary pattern as the group of reference. ^a^ MUFA = Mono Unsaturated Fatty Acid. ^b^ SFA = Saturated Fatty Acid. ^c^ DHA = Docosahexaenoic Acid.

**Table 3 nutrients-13-03884-t003:** Overview of suitable research items. (Arrows indicate an increase (↑), no change (→) or a decrease (↓)).

Authors	Participants	Training Status	Study Design	Nutritional Intervention	Exercise Intervention	Performance Measurements	Outcome and Direction of Outcome
Baguet et al. (2011)	Group 1 (*n* = 10) Age: 21.5 ± 1.7 years Group 2 (*n* = 10) Age: 20.8 ± 1.4 years	Physically active (2–3 h per week)	Intervention (5 weeks)	Group 1: Mixed diet Group 2: Lacto-ovo vegetarian diet	Sprint training (running and cycling) Week 1–2: 2× week Week 3–5: 3× week	Power output on an electromagnetically braked cycle ergometer	Mean power output: ↑ (Independent of groups)
Blanquaert et al. (2018)	Group 1 (*n* = 10) Age: 25.9 ± 9.0 years Group 2 (*n* = 15) Age: 25.4 ± 7.1 years Group 3 (*n* = 14) Age: 25.5 ± 6.6 years	-	Intervention (6 months)	Group 1: Omnivorous diet Group 2: Lacto-ovo vegetarian diet + placebo Group 3: Lacto-ovo vegetarian diet + β-alanine and creatine	-	VO_2_max (mL/kg/min) via an incremental cycling test	VO_2_max: → Body weight: → (Independent of groups)
Boutros et al. (2020)	*n* = 56 Age: 25.6 ± 4.1 years 28 vegan 28 omnivorous	150–200 min aerobic physical activity/week	Cross-sectional	-	-	Estimated VO_2_max (mL/kg/min) via cycle ergometer Muscle strength (1RM of leg and chest press)	Estimated VO_2_max in vegans: ↑ Muscle strength: → Body weight: →
Campbell et al. (1999)	Group 1 (*n* = 9) Age: 60 ± 1 years Group 2 (*n* = 10) Age: 58 ± 2 years	Sedentary	Intervention (12 weeks)	Group 1: Habitual omnivorous diet Group 2: Self-selected lacto-ovo-vegetarian diet	Resistance training (2×/week)	Dynamic muscular strength (1RM)	Dynamic muscular strength: ↑ (Independent of groups)
Haub et al. (2002)	Group 1 (*n* = 10) Age: 63 ± 3 years Group 2 (*n* = 11) Age: 67 ± 6 years	-	Intervention (12 weeks)	Group 1: Self-selected lacto- ovo-vegetarian diet supplemented with beef Group 2: Self-selected lacto- ovo-vegetarian diet supplemented with vegetable protein (soy)	Resistance training (3×/week)	Muscular strength of the lower and upper body	Lower body strength: ↑ (Independent of groups) Upper body strength: ↑ (Independent of groups)
Haub et al. (2005)	Group 1 (*n* = 10 Group 2 (*n* = 11) Age: 65 ± 5 years	-	Intervention (14 weeks)	Group 1: Self-selected lacto- ovo-vegetarian diet supplemented with beef Group 2: Self-selected lacto- ovo-vegetarian diet supplemented with vegetable protein (soy)	Resistance training (3×/week)	Muscular strength of the lower and upper body (Three maximum repetitions at 20%, 40%, 60% and 80% of the 1RM at the time of the testing	Lower body strength: ↑ (Independent of groups) Upper body strength: ↑ (Independent of groups)
Hevia-Larraín et al. (2021)	*n* = 38 19 vegan Age: 26 ± 5 years 19 omnivorous Age: 26 ± 4 years	physically active but not involved in resistance training for at least 1 year	Intervention (12 weeks)	-	Resistance training (2×/week)	Leg press 1RM	Lower body strength: ↑ (Independent of groups)
Hietavala et al. (2012)	*n* = 9 Age: 23.5 ± 3.4 years	Recreationally active	Intervention (18–24 days)	Group 1 (*n* = 5): (1.) 4 d habitual omnivorous diet (2.) 10–16 d wash-out phase (habitual omnivorous diet) (3.) 4 d low-protein vegetarian diet Group 2 (*n* = 4): (1.) 4 d low-protein vegetarian diet (2.) 10–16 d wash-out phase (habitual omnivorous diet) (3.) 4 d habitual omnivorous diet	-	VO_2_ (L/min) at 40%, 60% and 80% of VO_2_max VO_2_max	After low-protein vegetarian diet: VO_2_ ↑ (at 40%, 60% and 80% of VO_2_max)
Kròl et al. (2020)	*n* = 52 22 vegan Age: 32 ± 5 years 30 omnivorous Age: 30 ± 5 years	Physically active (at least 3×/week)	Cross-sectional	-	-	Peak power output (W) VO_2_max (mL/kg/min)	VO_2_max in vegans: ↑ Peak power output: → Body weight in vegans: ↓
Lynch et al. (2016)	*n* = 70 27 vegetarian 43 omnivorous Age: 21–58 years	Competitive club sports team	Cross-sectional	-	-	VO_2_max (mL/kg/min) Peak torque leg extension	VO_2_max (mL/kg/min) max in female vegetarians: ↑ VO_2_max (L/min): → Body weight in female vegetarians: ↑ (n.s.)
Nebl et al. (2019)	*n* = 74 26 omnivorous 24 lacto-ovo vegetarian 24 vegan Age: 18-35 years	Recreational runners	Cross-sectional	-	-	Maximum exercise capacity (Pmax/bodyweight) Power output related to lean body mass (Pmax/LBM)	Maximum exercise capacity: → Power output related to lean body mass: →
Page et al. (2021)	*n* = 25 16omnivorous Age: 21 ± 1 years 9 vegan Age: 24 ± 3 years	No history of resistance or endurance exercise training in the preceding six months	Cross-sectional	-	-	VO_2_max (ml/kg/min) and (L/min) Maximal voluntary isometric contraction (MVIC) force	VO_2_max: → MVIC: →
Veleba et al. (2016)	Group 1 (*n* = 7) Age: 57.7 ± 4.9 years Group 2 (*n* = 37) Age: 54.6 ± 7.8 years	-	Intervention (12 weeks)	Group 1: Hypocaloric (−500 kcal) conventional diet Group 2: Hypocaloric (−500 kcal) vegetarian diet	Aerobic exercise 3×/week	Maximum performance (Watt_max_) VO_2_max (ml/kg/min)	Group 1: Maximum performance: → VO_2_max: → Group 2: Maximum performance: ↑ VO_2_max: ↑
Wells et al. (2003)	Group 1 (*n* = 10) Group 2 (*n* = 11) Age: 59-78 years	-	Intervention (12 weeks)	Group 1: Self-selected lacto-ovo vegetarian diet + beef protein supplement (0.6 g/kg/day) Group 2: Self-selected lacto-ovo-vegetarian diet + vegan protein supplement (0.6 g/kg/day)	Resistance training (3×/week)	Maximal strength (1RM)	Baseline maximal strength: → Maximal strength after 12 weeks of resistance training: ↑ (independent of group) Strength in knee extension in Group 2 compared to Group 1: ↑

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
