# Peer review of "The Impact of Vegan and Vegetarian Diets on Physical Performance and Molecular Signaling in Skeletal Muscle"

_nutrients, 2021, doi:10.3390/nu13113884_

Round 1

Reviewer 1 Report

The topic is very interesting, and despite the few papers in the literature, the manuscript extensively examines both aspects of endurance and strength physical performance and comparisons between different types of diets.

Although the individual articles included are described in detail in the text, I think the section regarding methods should be structured more systematically in the same chapter. What is the type of study design in the selected articles (cross-sectional, longitudinal, intervention)? What are the characteristics of the population? What is the age group analyzed?
When the authors talk about vitamin D fluctuations in response to different dietary patterns, they should specify how greater adherence to the Mediterranean diet, which is a plant-based model, was found to be strongly associated with better circulating levels of this vitamin, demonstrating the positive aspect of this dietary pattern (10.1080/09637486.2020.1744533).

Line 477 paragraph 6.1. Proteins and amino acids and their impact on molecular signaling. Mention the data on nutritional imbalance, especially protein, found in aging adults and involved in delineating the nutritional frailty phenotype in the elderly. (10.1016/j.arr.2020.101148 and 10.1111/joim.13384). 
I find this narrative review very informative, with many references.
I appreciate the figures that clarify the concepts expressed in the text.
The authors should improve on the above points.

Author Response

Comment #1

This topic is very interesting, and despite the few papers in the literature, the manuscript extensively examines both aspects of endurance and strength physical performance and comparisons between different types of diets.

            Reply: We thank the Reviewer for this motivating comment on our manuscript.

Comment #2

Although the individual articles included are described in detail in the text, I think the section regarding methods should be structured more systematically in the same chapter. What is the type of study design in the selected articles (cross-sectional, longitudinal, intervention)? What are the characteristics of the population? What is the age group analyzed?

Reply: We thank You for Your comment. We changed the term comparative to cross-sectional in the text. In order to not disturb the flow of reading, we have made additions to the overview table (Table 3). Therefore, we changed the column Duration to Design, added the age group to the column Participants and added the column training status.

Comment #3

When the authors talk about vitamin D fluctuations in response to different dietary patterns, they should specify how greater adherence to the Mediterranean diet, which is a plant-based model, was found to be strongly associated with better circulating levels of this vitamin, demonstrating the positive aspect of this dietary pattern

Reply: We thank You for Your comment. We specified the vitamin D level aspects concerning to Your comment on page 2, lines 68ff.       

Comment #4

Line 477 paragraph 6.1. Proteins and amino acids and their impact on molecular signaling. Mention the data on nutritional imbalance, especially protein, found in aging adults and involved in delineating the nutritional frailty phenotype in the elderly.

Reply: We thank You for Your comment. We mentioned this aspect on page 20, lines 506ff.

Reviewer 2 Report

This thoroughly conducted systematic review shows important effects of three nutritional lifestyles on muscular metabolism and functionality. I do have some remarks:

  • Apart from isometric and isokinetic exercises, does there exist a difference between omnivore, vegetarian or vegan diets, when looking at isotonic exercises?
  • The vegan diet yields considerable heterogeneity, with the omittance of non-rooted vegetables being a classic variation, and the authors should make note of this.
  • Concerning amino acid metabolism, it should be noted that carnosine and taurine, both especially present in red meat, could have beneficial effects in this setting (as compared to vegetarian and vegan), and the authors should comment on this.
  • Despite vitamin D also being needed in (skeletal) muscular metabolism, a positive effect of vitamin D suppletion has never been demonstrated in several meta-analyses (in any setting at all, by the way, unless for newborns receiving breastmilk), and the authors should comment briefly on this.

Author Response

Comment #1

This thoroughly conducted systematic review shows important effects of three nutritional lifestyles on muscular metabolism and functionality.

            Reply: We thank the Reviewer for this appreciative comment on our manuscript.

Comment #2

Apart from isometric and isokinetic exercises, does there exist a difference between omnivore, vegetarian or vegan diets, when looking at isotonic exercises?

Reply: Thank You for Your comment. Chapter 5.4 Evidences on vegetarian and vegan diets and strength performance studies also used isotonic strength diagnostics. In general, when diagnostics are not specified as isokinetic or isometric, strength was assessed isotonically. For clarification, we added the term isotonic to the description at the appropriate place (see lines 344, 367, 386, 395, 416).

Comment #3

The vegan diet yields considerable heterogeneity, with the omittance of non-rooted vegetables being a classic variation, and the authors should make note of this.

Reply: Thank You for Your comment. After reviewing the literature, we have not found a source for the non-rooted vegetable-omitting variation, but we made note of this distinction on page 2, lines 45ff.

Comment #4

Concerning amino acid metabolism, it should be noted that carnosine and taurine, both especially present in red meat, could have beneficial effects on this setting (as compared to vegetarian and vegan), and the authors should comment on this.

Reply: Thank You for Your comment. We covered the role of carnosine in endurance performance on page 10, lines 49ff.

We mentioned the role of taurine in molecular signaling on page 23, lines 631ff.

For better comprehension, we also included the explanation of AMPK from chapter 7 into this paragraph.

Additionally we added the role of taurine in molecular signaling to Figure 2 and adjusted the figure description (see lines 463ff.)

Comment #5

Despite vitamin D also being needed in (skeletal) muscular metabolism, a positive effect of vitamin D suppletion has never been demonstrated in several meta-analyses (in any setting at all, by the way, unless for newborns receiving breastmilk), and the authors should comment briefly on this.

            Reply: We thank the reviewer for the response. Indeed the general assumption of positive effects of vitamin d supplementation on exercise performance is inconsistent. While some studies show positive effects and others not we specified that issue in the endurance performance part (page 10, lines 44ff.) and the strength performance part (page 17, lines 336f.).

Round 2

Reviewer 1 Report

The manuscript sounds good now. Thank you for having addressed all suggestions.